# Interactions between an Associative Amphiphilic Block Polyelectrolyte and Surfactants in Water: Effect of Charge Type on Solution Properties and Aggregation

**DOI:** 10.3390/polym13111729

**Published:** 2021-05-25

**Authors:** Patrizio Raffa

**Affiliations:** Department of Chemical Engineering, Faculty of Science and Engineering, University of Groningen, Nijenborgh 4, 9747 AG Groningen, The Netherlands; p.raffa@rug.nl

**Keywords:** amphiphilic block copolymers, polyelectrolytes, surfactants, polyelectrolyte-surfactant complexes, rheology, surface activity

## Abstract

The study of interactions between polyelectrolytes (PE) and surfactants is of great interest for both fundamental and applied research. These mixtures can represent, for example, models of self-assembly and molecular organization in biological systems, but they are also relevant in industrial applications. Amphiphilic block polyelectrolytes represent an interesting class of PE, but their interactions with surfactants have not been extensively explored so far, most studies being restricted to non-associating PE. In this work, interactions between an anionic amphiphilic triblock polyelectrolyte and different types of surfactants bearing respectively negative, positive and no charge, are investigated via surface tension and solution rheology measurements for the first time. It is evidenced that the surfactants have different effects on viscosity and surface tension, depending on their charge type. Micellization of the surfactant is affected by the presence of the polymer in all cases; shear viscosity of polymer solutions decreases in presence of the same charge or nonionic surfactants, while the opposite charge surfactant causes precipitation. This study highlights the importance of the charge type, and the role of the associating hydrophobic block in the PE structure, on the solution behavior of the mixtures. Moreover, a possible interaction model is proposed, based on the obtained data.

## 1. Introduction

Polyelectrolyte surfactant complexes (PESCs) have been extensively studied over the last few decades due to their interesting physico-chemical properties, leading to important applications in colloid science such as detergency, flocculation, formulation of pharmaceuticals, cosmetics, paints and coating, and enhanced oil recovery [1,2,3,4,5,6,7,8]. Moreover, the study of interactions between polyelectrolytes (PE) and surfactants can serve as a simple model for biological processes, as the most important biopolymers (proteins, RNA, DNA) are in fact polyelectrolytes, and cell membranes are made of self-assembled surfactant structures (micelles, vesicles, lamellae). The complex behavior of PE in solution arises from electrostatic and hydrophobic interactions that lead to a variety of possible conformations and self-assembled structures with significant impact on the final properties of the mixture, in particular of the rheological and interfacial varieties.

### 1.1. Models of PE in Solution

In order to better describe the interactions between polyelectrolytes (PE), surfactants and their resulting solution properties, the general models used to describe PE in solution are first shortly summarized here, based on excellent reviews already published on the subject [9,10,11,12]. Isolated PE chains have been historically described starting from the Flory theory, adding a term for electrostatic interactions to the free energy of a Gaussian chain (Equation (1)) [10].
(1)$$\frac{U\left(r\right)}{k_{B}T}=\frac{3}{2l^{2}}\underset{i}{\sum }{\left(r_{i+1}-r_{i}\right)}^{2}+\underset{i}{\sum }\underset{j<i}{\sum }l_{B}\frac{z_{i}z_{j}}{r_{ij}}e^{-\kappa r_{ij}}+\frac{U_{LJ}\left(r_{ij}\right)}{k_{B}T}$$
where *U(r)* is the potential energy as a function of the position of the monomers, *k_B_* the Boltzmann constant, *T* the temperature, *l* the monomer size, *l_B_* the Bjerrum length, *κ* the inverse of the Debye length, *z* the charge multiplicity of the monomer, and *U_LJ_* is the repulsive potential, typically in the Lennard-Jones form. It can be noticed that the electrostatic interaction is expressed as the Debye–Hückel screened potential, to account for the presence of other electrolytes.

Models based on Equation (1) describe the PE as a stiff, semi-flexible chain, fully extended in salt-free conditions (Figure 1a). As external electrolytes are introduced, the PE chain stiffness decreases, and the polymer assumes conformations more similar to an expanded self-avoiding chain in good solvent (Figure 1b) [9]. This is due to increased charge screening and reduced osmotic effects from the PE counterions. To account for experimental observations that the electrostatic interactions in PE chains has a longer range than the Debye length [13], other models have been introduced, based on the blob theory of De Gennes [14]. The blob size, determined by an electrostatic persistence length $l_{p}^{el}$ (Equation (2)), can increase or decrease as a function of salinity, altering the chain stiffness [9,13].
(2)$$l_{p}^{el}=l_{p}^{0}+\frac{l_{B}\alpha^{2}}{4{\left(\kappa l\right)}^{2}}$$

In Equation (2), α is the degree of ionization and $l_{p}^{0}$ the persistent length of the unperturbed blob. As in the previous description, also here the two characteristic lengths of electrostatic interactions (Bjerrum and Debye lengths) play a relevant role.

For hydrophobic PE, the balance between electrostatic and hydrophobic interactions results in various possibilities. Depending on the hydrophobicity, the PE will assume a so-called “pearl-necklace” conformation (Figure 1c), characterized by connected hydrophobic beads, which size is governed by Rayleigh stability [10,15,16]. It should be also observed that in semi-dilute or concentrate regime, intermolecular associative behavior occurs, with formation of transient networks and strong consequences on the solution rheology. This has been extensively exploited in applications such as enhanced oil recovery [17,18]. As the hydrophobicity increases, coil-to-globule transition and aggregation will start occurring (Figure 1d), resulting in complex structures and solution behavior. Highly hydrophobic PE with well-defined polymer architectures have been mostly studied experimentally by DLS, microscopy, rheology and surface tension measurements [19,20,21].

The distribution of hydrophobic groups and charges on the PE also plays a fundamental role in their solution behavior [22,23]. When hydrophobic and charged groups are completely segregated in block structures, formation of supra-molecular micellar aggregates will occur in solution.

Amphiphilic block polyelectrolytes (ABPE) are a very important class of PE. Due to their ability to self-assemble, they have very interesting solution behavior, and they have been extensively studied, both theoretically and experimentally, and used for many applications [6,23,24,25,26,27,28]. Scaling laws for these systems have been obtained by adapting mean-field theories developed for micellization of block copolymers and PE brushes on surfaces [29,30,31,32]. The properties are very dependent on the polymer composition and the type of self-assembled structures formed in solution. For spherical star-like micelles (PE block much longer than hydrophobic block), scaling laws very similar to those obtained for general PE have been derived, with the polyelectrolyte corona extending in solution or contracting, depending on salinity (Figure 1e,f, respectively) [29].

### 1.2. Studies on PESCs

In presence of surfactants, formation of PESCs occur, with PE conformational changes ultimately leading to interesting solution behavior, which is relevant for most of the intended applications [1]. The majority of published research on PESCs, either experimental or modeling studies, considers opposite sign systems [2,33,34,35,36,37,38,39,40,41,42,43,44,45]. In this case a strongly cooperative mechanism exists, where electrostatic attraction causes the surfactant micellization process to happen at a much lower concentration than the CMC [33,38,46]. The structure formed above the so-called critical aggregation concentration (CAC) are polymeric chains wrapped around surfactant micelles (analogous to the pearl-necklace model presented above) [2,46,47]. When a point of charge neutralization is reached, precipitation or separation of a PESC-rich phase typically occurs [1,39,40,42] that may eventually redissolve by adding excess surfactant [48]. 

For same sign systems, less studies have been performed, but it is generally concluded both from experiments and simulations [49,50,51,52] that hydrophobic interactions overcome the electrostatic repulsion between charges, causing the surfactant to co-micellize with the polyelectrolyte (Figure 2). Also, interactions with nonionic surfactants have been subject of relatively few publications. Often, they are studied in combination with charged surfactants, to investigate competitive or synergistic effects [53,54,55,56,57].

A very limited number of studies focus on interactions of ABPE with surfactants [58,59,60,61,62,63]. In these studies it has been evidenced by various techniques that the presence of surfactants alters the shape and size of aggregates (usually spherical micelles) formed by the ABPE. Analogous behavior has been observed for gradient amphiphilic copolymers [64]. However, the impact of these structural changes on solution properties has not been systematically studied, especially concerning surface activity and solution rheology.

### 1.3. Aim of the Work

In this work, the interactions of an ABPE with different surfactants (anionic, cationic, and nonionic) have been investigated in water solution via surface tension and shear viscosity measurement for the first time. The polymer studied here is a triblock PMAA_312_-b-PS_108_-b-PMAA_312_ synthesized in previous work by sequential ATRP [65] and designed primarily as viscosifier for EOR [28]. The study of its interaction with surfactants is relevant for this field [66,67]. Models of interaction are proposed on the basis of the known theory and experimental observations. This study intends to provide insights into the behavior of amphiphilic block polyelectrolytes in presence of variously charged surfactants, from rheological and interfacial point of view, that are of importance for several of the mentioned applications.

## 2. Materials and Methods

The ABPE studied in this work, a triblock polystyrene-b-poly(sodium methacrylate) PMAA_312_-b-PS_108_-b-PMAA_312_, was synthesized and fully characterized in previous work (>95% form NMR analysis, average Mn (sodium salt) = 64,900 g/mol) [65].

Enordet J11111, an alkyl ether sulfonate anionic surfactant (active matter > 95%, average Mn = 910 g/mol), was kindly provided by Shell Global Solution International B.V. (The Hague, Netherlands) and used as received. Cetyltrimethylammonium Bromide (≥98%), Poly(ethylene glycol) methyl ether (flakes, average Mn = 2000 g/mol) and Pluronic P123 (triblock PEO-PPO-PEO, average Mn = 5800 g/mol, approx. EO % = 30) were purchased from Sigma-Aldrich (Zwijndrecht, Netherlands) and used as received.

Solutions were prepared in MilliQ water. Typically, the polymer was dissolved first in water at the desired concentration. Homogeneous solutions were obtained by overnight stirring. Then the required amount of surfactant was added, and the solution stirred for additional 2 h before taking the measurements.

Shear viscosity measurements were performed in a Haake Mars III rotational rheometer (Thermo Scientific, Waltham, MA, USA) with cone and plate geometry, at room temperature. Each measurement was performed in duplo.

Surface tension measurements were performed with a pendant drop OCA 15EC tensiometer from Dataphysics (Filderstadt, Germany), at room temperature. The surface tension values were obtained as average of at least 5 measurements. 

Measurements repeated at a distance of several days gave consistent results.

## 3. Results

### 3.1. Studied Systems

For this study, four surfactants with different characteristics have been used (Table 1). The main purpose was to investigate the interactions in solution with the micellar aggregates formed by an amphiphilic triblock PMAA_312_-b-PS_108_-b-PMAA_312_ copolymer, as a function of the different charge present on the surfactant. An anionic, a cationic and two nonionic surfactants have been chosen. 

The surfactants were characterized by surface tension measurements in demineralized water (see Appendix A). The polymer was synthesized in previous work [65] via sequential ATRP of styrene using a difunctional initiator, followed by chain extension with tert-butyl methacrylate, and hydrolysis of the product to obtain the acid form. This was completely neutralized with NaOH and dialyzed to remove the excess of base. When dissolved in water, the solution reaches a pH of 9.5, indicating complete neutralization [65]. GPC traces are reported in the Appendix A. This polymer is known to form stable spherical micellar aggregates, with a kinetically frozen hydrophobic core [60,65], therefore it does not have a CMC.

#### 3.1.1. Same Charge

Figure 3a shows the surface tension (γ) curve for the anionic surfactant Enordet at increasing concentration of polymer in demineralized water.

It is evidenced that the CMC of the surfactant shifts to higher values as the concentration of the polymer increases. Similar behavior has been observed for PBA-b-PAA copolymers mixtures with SDS [60], and it is common for PESCs in general [4,8]. For low surfactant/polymer concentration ratio, the surface tension decreases less than for the pure surfactant. These two observations suggest that the surfactants molecules adsorption on the polymer compete with adsorption at the air/water interface. As the surfactant concentration increases, it eventually reaches saturation, achieving the same value of surface tension of the surfactant alone. Interestingly, if the surface tension is plotted versus the molar surfactant/polymer ratio, making it independent on the polymer concentration, a master curve is obtained (Figure 3b). This curve shows that adsorption of the surfactant at the air/water interface occurs predominantly when the concentration of surfactant is between 1 and 10 times that of the polymer. These data do not give information about the nature of the aggregates formed in solution, but they show that the ABPE interacts with same-charge surfactants already at low concentration of surfactant. This happens most likely via hydrophobic interactions, analogously to other amphiphilic polyelectrolytes [49,50]. In the case of the ABPE, this means that the surfactant interacts with the hydrophobic core of the micellar aggregate [60]. 

#### 3.1.2. Opposite Charge

Surface tension plot of CTAB in water and presence of 0.5 weight % of ABPE are qualitatively similar to Figure 3, and are reported in the Appendix A. A shift in the CMC of the surfactant is observed also in this case, but the most notable phenomena is that for CTAB concentration above 3 mM the solution becomes milky and much less viscous, suggesting the precipitation of macroscopic aggregates. This corresponds to about 12% degree of neutralization. This value is lower than that reported for PESCs of Poly(sodium acrylate) CTAB complexes previously studied (above 40%) [34], which is not surprising, as the ABPE investigated here is much more hydrophobic, therefore precipitation can be expected to occur at lower degree of neutralization. It can also be noticed that precipitation occurs when the CTAB concentration is above the apparent CMC (Appendix A), suggesting that the saturation limit has been already reached. 

#### 3.1.3. Nonionic Surfactants

As shown in Appendix A, the presence of ABPE has on Pluronic the same effect as the charged surfactants, shifting the apparent CMC at higher concentration. For PEGMe a different behavior was observed. The surface tension plot in presence of polymer does not change significantly from that of pure surfactant, but contrarily to what happens with the previous mixtures, the surface tension slightly decreases (Appendix A). Considering that the polymer itself is virtually non-surface active [68], this can be described as a synergistic effect, not displayed by the other combinations studied here. The decrease in surface tension suggests that the low molecular weight molecule, although not highly surface active on itself, helps the ABPE to unfold and adsorb at the air/water interface [4,8].

### 3.2. Solution Rheology

Solution rheology of PESCs is very important for several applications, such as enhanced oil recovery [66,69] or foam and emulsion stabilization [70], and it can give additional information on the nature of the aggregates formed. For ABPE/surfactant mixture no rheological studies have been previously reported to the best of my knowledge. Figure 4a reports shear viscosity for 1 wt % solution of the amphiphilic block polyelectrolyte (overlapping regime [32,65]), at increasing concentration of Enordet. The molar surfactant/polymer ratio goes approximately from 1 to 5.

The amphiphilic block polyelectrolyte investigated here it is known to give highly viscous solutions at relatively low concentration in demineralized water, with strong non-Newtonian effects (shear thinning) [65]. This is usually interpreted as the result of interacting large micellar aggregates, as the ones represented in Figure 1f.

It is here observed that when the surfactant is added, the viscosity starts decreasing as the concentration of surfactant exceeds that of the polymer. The effect of a large excess of surfactant has also been investigated, for polymer solutions at 0.1 wt % concentration (Appendix A). In this semi-dilute regime, the polymer solution still behaves as a non-Newtonian fluid. The viscosity keeps decreasing as the surfactant concentration is increased, and it seem to approach a limit for a large excess of surfactant (>100 times), where the viscosity remains however significantly higher than that of water, and still showing non-Newtonian shear thinning behavior. The decrease in viscosity can be in principle explained by two separate effects: the disruption of the ABPE micellar aggregates to a smaller size, and the shrinking of the PE corona due to a screening effect from the charges introduced in solution by the surfactant. These data support the hypothesis made for analogous APBE [60] that the surfactant is able to dissolve the kinetically frozen core of the polymer aggregate.

Finally, Figure 4b reports the shear viscosity for 1 wt % solutions of polymer in water and in presence of 0.1 wt % of the other three surfactants. Interestingly, the cationic surfactant does not affect significantly the viscosity, up to the point where the complex starts precipitating (3 mM concentration). Above this value of CTAB concentration, the viscosity drastically decreases (not measured due to the heterogeneous nature of the formed mixture).

The nonionic surfactants reduce the viscosity already at much lower molar concentration than CTAB. Pluronic, which is the surfactant with the lowest CMC of those investigated, seems to have the most pronounced effect in viscosity reduction, while PEGMe has a comparable effect to that of Enordet, even though it is not much surface active. It is worth noticing that as both Pluronic and PEGMe are nonionic surfactants, the viscosity reduction is certainly not due to a salt effect in these cases. 

### 3.3. Proposed Model

Based on all the experimental observation shown in the previous section, and information from literature on analogous systems [60,64], a qualitative model for interaction between ABPE and surfactants of various charge is presented here and illustrated in Figure 5.

Surface tension measurements suggest that all surfactants interact with the ABPE aggregates, regardless of their charge. The shift in CMC indicates that complexations with the polymer compete with adsorption at the surface and with formation of free micelles, as it is commonly reported for other kind of PESCs [1,4,8]. We can notice that nonionic and anionic (same charge) surfactants show similar behavior: increase in apparent CMC and decrease in viscosity, starting from relatively low concentration. A model compatible with these findings, with previous models [60,64] and with computer simulation on same-charge PESCs [51,52] is illustrated in Figure 5. The surfactant interacts primarily via hydrophobic groups, forming mixed micelles. These have smaller size compared to the initial ones (due to lower aggregation number), causing a decrease in viscosity. For the anionic surfactant, this may be combined with a salt effect [65], also causing contraction of the PE arms.

For the cationic surfactant, the observed behavior is different. The most notable differences are the formation of a precipitate above a certain surfactant concentration, and a relatively small effect on viscosity before that point is reached. Based on this, the model illustrated in Figure 5b is proposed. In this case, the interaction is prevalently electrostatic, therefore the hydrophobic core is not touched, and the viscosity is not much affected. The surfactant will still micellize, probably at lower apparent CMC due to cooperative binding to the polyelectrolytic arms. At high enough concentration, a precipitate analogous to already known PESCs of opposite sign starts forming. This happens at a relatively low neutralization ratio (around 10%), probably due to the already big aggregates formed by the ABPE. All system investigated show reduction of surface tension, therefore adsorption at the air/water interface still occurs. This could be ascribed to the surfactant alone, but the observations made for PEGMe (Appendix A), and literature [4,8] suggest that the ABPE/surfactant complexes may also adsorb. Other experiments such as DLS, zeta potential, neutron reflectivity, surface rheology, or SAXS, could help confirming this model.

## 4. Conclusions

In this work, the interactions in water of an anionic amphiphilic triblock PMAA-b-PS-b-PMAA polyelectrolyte with surfactants bearing same charge, opposite charge, and no charge respectively, have been investigated by surface tension and shear viscosity measurements, for the first time.

The knowledge of solution rheology and surface activity of these mixtures is relevant for many applications, including enhanced oil recovery, foam and emulsion stabilization, flocculation, and formulation of paints, coatings and pharmaceutical products [1,2,3,4,5,6,7,8]. As the investigation of PESCs is usually limited to non-aggregating PE, and only sparse experimental [58,59,60,64] and modeling [61,63] studies have been performed for ABPE, this work aimed at providing new insight on the behavior of such systems.

In the majority of the investigated cases (Enordet, CTAB and Pluronic), the surface activity decreases and the CMC of the surfactant shifted to higher values, confirming the general behavior of PE [1]. Regardless of the charge type, it is confirmed that the surfactant interacts with the polymer aggregates. An interesting synergistic effect was observed for PEGMe, a nonionic weak surfactant, where the mixtures possess lower surface tension values than the single components. This behavior is well-known for typical PE [8], but it was apparently never reported for ABPE.

The shear viscosity of the mixtures was investigated for the first time for ABPE/surfactant systems. The solution behavior remains qualitatively similar to that of the pristine polymer (non-Newtonian, shear thinning), but the absolute values of viscosity up to an order of magnitude for mixtures with anionic and nonionic surfactants. In the case of CTAB (opposite charge system), the viscosity is not greatly affected at low surfactant concentration, but a precipitate starts forming when the degree of neutralization reaches values above 10%. These results are relevant for applications of these mixtures as rheology modifiers.

Based on these experiments, a simple model of aggregation was proposed here, which is consistent with previous models presented for analogous amphiphilic polymers [60]: same charge or nonionic surfactants seem to be able to dissolve the hydrophobic core, while cationic ones interact primarily via electrostatic attraction, causing precipitation of larger aggregates. This was never reported before for ABPE, although it is a typical observation for other PE [5,36,46,47].

In conclusion, this study provides new insight on the behavior of ABPE aggregates with surfactants of various nature, confirming previous models [60,64], but also proposing a new inclusive one, and providing new quantitative data on surface activity and rheological behavior of such mixtures, which are also relevant for practical applications.

Additional experiments aimed at establishing the nature of aggregates such as DLS, zeta-potential, SAXS, cryo-EM, and neutron reflectivity would be very interesting, and may contribute to confirming the proposed interaction mechanism.

## Figures and Tables

**Figure 1 polymers-13-01729-f001:**
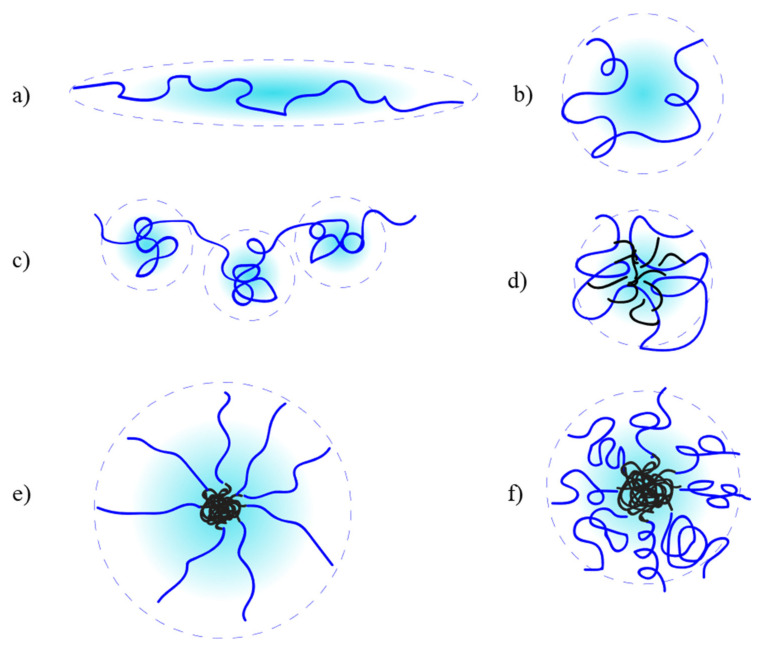
Conformation of various kind of PE in solution: (**a**) salt-free conditions; (**b**) in saline water; (**c**) pearl-necklace model for hydrophobic PE; (**d**) globule conformation for highly hydrophobic PE; (**e**) amphiphilic block polyelectrolytes (ABPE) in salt-free conditions; (**f**) APBE in saline water.

**Figure 2 polymers-13-01729-f002:**
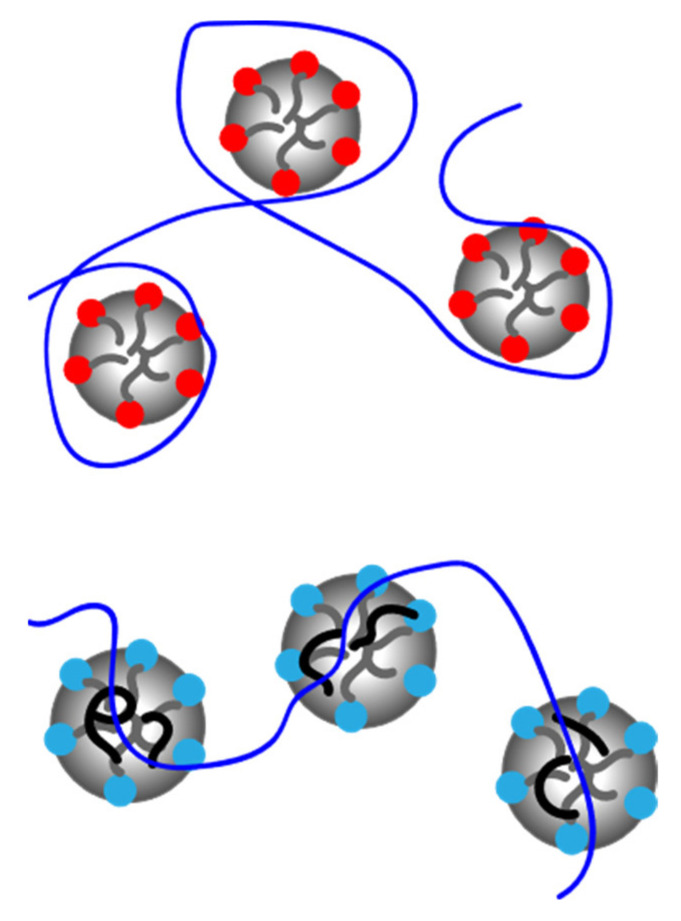
Graphical representation of PESCs of opposite sign charge (**top**) and same sign charge (**bottom**).

**Figure 3 polymers-13-01729-f003:**
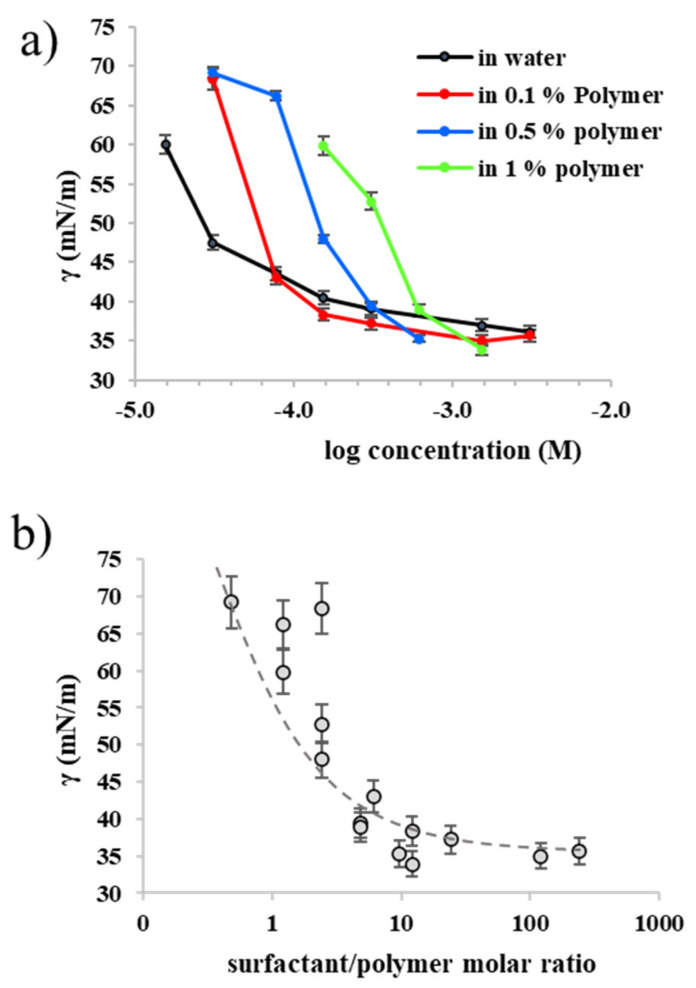
(**a**) surface tension curve of Enordet at different polymer concentrations; (**b**) master curve of surface tension as a function of the surfactant/polymer ratio. The dotted line is intended as a guide for the eye.

**Figure 4 polymers-13-01729-f004:**
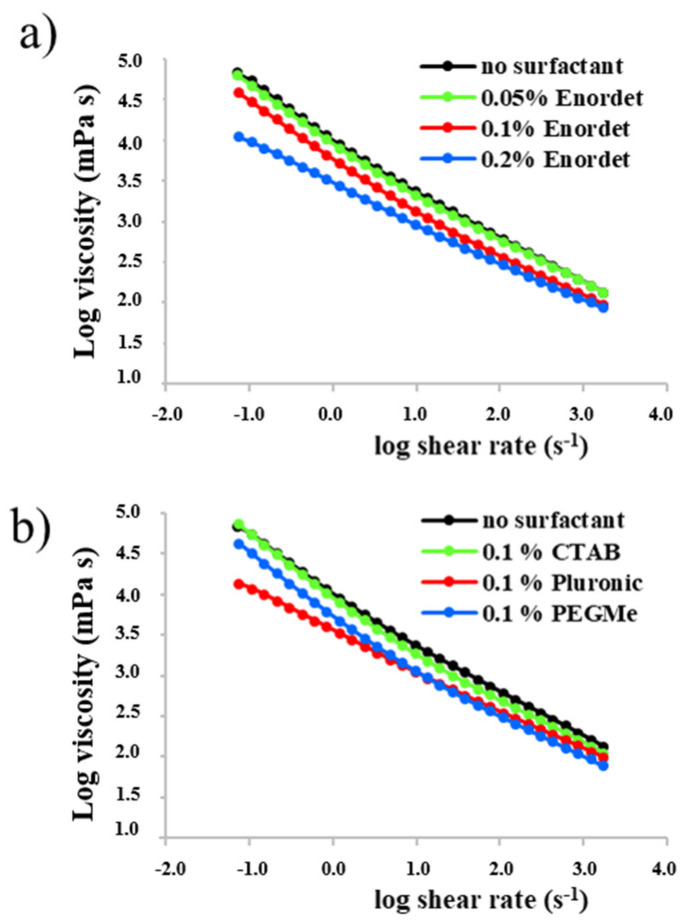
Shear viscosity of a 1 wt % polymer solution with (**a**) increasing amount of Enordet; (**b**) 0.1% of CTAB, Pluronic and PEGMe.

**Figure 5 polymers-13-01729-f005:**
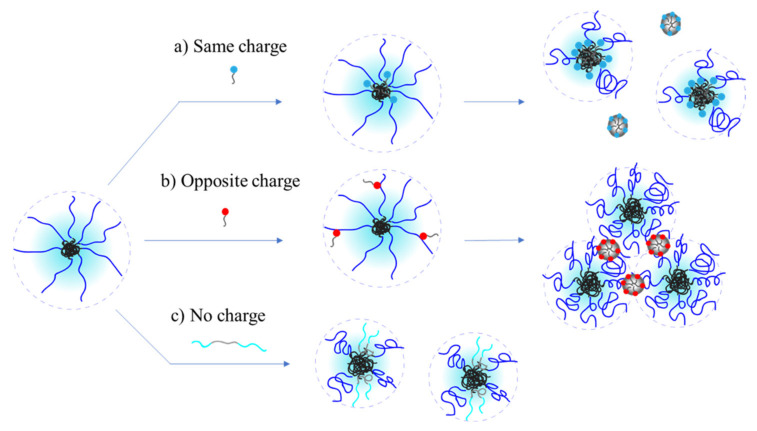
Proposed schematic model for APBE/surfactant interactions.

**Table 1 polymers-13-01729-t001:** Structure of the surfactants employed in this study.

Surfactant	Structure
Enordet J1111	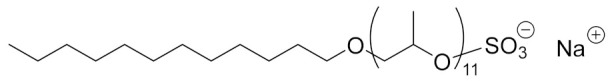
CTAB	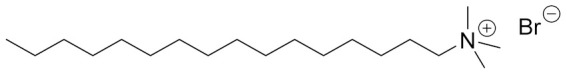
PEGMe	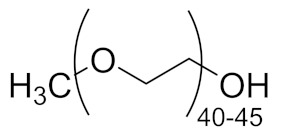
Pluronic P123	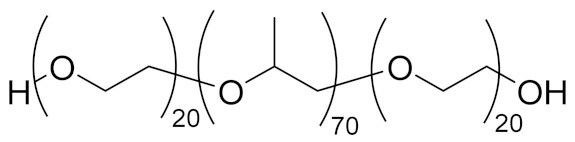

## Data Availability

The raw data presented in this study are available on request from the corresponding author.

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
