# Peer review of "Interactions between an Associative Amphiphilic Block Polyelectrolyte and Surfactants in Water: Effect of Charge Type on Solution Properties and Aggregation"

_polymers, 2021, doi:10.3390/polym13111729_

Round 1

Reviewer 1 Report

Recommendation: This paper represents some new contribution and is publishable.

Comments: “This article elucidated the systematic investigation of interactions between interactions between an anionic amphiphilic triblock polyelectrolyte and different types of surfactants which is of great interest for both fundamental and applied research. The effects of surfactant on viscosity and surface tension of amphiphilic triblock polyelectrolyte solutions were detailly studied. Moreover, a possible interaction model is proposed to base on obtained data. These investigations have important values. The paper is well organized and presented. Therefore, I would like to recommend it for publication in polymers.”

Author Response

I thank the reviewer for revising this work. I don't find in the reviewer's report particular points to respond to. I made a general revision of the manuscript, clarifying a few more details (see new version with track changes)

Reviewer 2 Report

polymers-1238404

The article: “Interactions Between an Associative Amphiphilic Block Polyelectrolyte and Surfactants in Water: Effect of Charge Type on Solution Properties and Aggregation” by Patrizio Raffa describes a study of the interactions between an anionic amphiphilic triblock polyelectrolyte (PMAA-b-PS-b-PMAA), synthesized by ATRP, and different types of surfactants bearing respectively negative, positive and no charge. These systems were investigated via surface tension and solution rheology measurements. Moreover, a possible interaction model is proposed, based on the obtained data.

In general the MS is well-written and organized in sections. The studies were well-performed and the explanations given by the author are well-documented and illustrated. I suggest that this work could be published in Polymers. Only a few minor comments were raised by reading the manuscript:

  • Concerning the homogeneity of the text, the author should correct the font size in page 2-4. It is different compared to the rest of the text that follows the template of the journal.
  • The author mentioned that the triblock copolymer that was used in this study was previously synthesized by ATRP and characterized by GPC and H-NMR according to Ref. 65 (Macromolecules, 2013, 46, 7106-7111). I checked the given reference but the molecular characteristics do not match with the ones that the author mentions in the text (PMAA312-PS108-PMAA312). Moreover in the reference the authors don’t provide a GPC chromatograph (only H-NMR is given). Could the author include the GPC chromatograph and the H-NMR spectrum of this specific sample (maybe in the SI)?
  • How the author calculated the total molecular weight of this triblock terpolymer?

Author Response

I thank the reviewer for the positive comments. About the point-to point answer:

  • I corrected the format of pages 2-4 according to the style of the rest of the manuscript.
  •  the polymer composiotion do matches the one given in the cited reference 65 (it can be found in scheme 1 and table S2). I noticed that the GPC traces were not reported in the previous paper, so I added them here in the newly submitted supplementary information file. To molecular weight determination was based both on GPC and H-NMR data. It should be noted that the molecular weight obtained by GPC is the one of the polymer precursor, before hydrolysis of the tert-butyl group, while the one reported here is calculated for the hydrolysied polymer in its sodium salt form.